# Design Method and Application of Stope Structure Parameters in Deep Metal Mines Based on an Improved Stability Graph

## Xingdong Zhao and Xin Zhou *

Laboratory for Safe Mining in Deep Metal Mine, Northeastern University, Shenyang 110819, China
* Correspondence: 2010397@stu.neu.edu.cn

**Abstract:** Deep mining has become an inevitable trend of mining development. Previously conducted studies have established that reasonable stope structure parameters are the premise to ensure the safe and efficient production of deep mines. In order to ensure the safety of deep mining, in this paper, we systematically review the existing stope structure parameter design methods, and then put forward a deep stope structure design method based on the stability of mining rock mass. Based on rock mass quality classification, this method uses a critical span graph and an improved stability graph, and fully considers the influence of joint occurrence and mining stress on the stability of surrounding rock, to design the stope structural parameters. Taking into consideration the deterioration of the quality of deep rock mass, we collect mining data at home and abroad, improve the stability graph, and make it suitable for the design of stope structural parameters with different mining methods. The design process of stope structural parameters is expounded through field engineering cases, and it has specific guiding significance for the design of stope structural parameters in deep metal mines.

**Keywords:** safety mining; deep mining; stope structural parameters; improved stability graph; engineering application

## 1. Introduction

With the gradual depletion of shallow metal mineral resources, the development of deep mineral resources has become an inevitable trend. After entering deep mining, the in situ stress increases and the mining technical conditions and environment deteriorate seriously, which presents significant challenges to the safety production of deep metal mines [1–3]. Reasonable stope structure parameters are an effective means to ensure safe and efficient production in deep metal mines.

At present, the design methods of stope structural parameters include the engineering analogy method, theoretical analysis method, numerical analysis method, and comprehensive analysis method; the latter two methods are widely used. Qiu et al. analyzed the stope stability of the Bainiuchang Mine based on the pillar area bearing theory, and they optimized the stope structure parameters using ANSYS numerical simulation [4]. Based on precision finite element modeling and simulation, Zhang et al. determined a reasonable width range and interval value of the strip for strip mining with subsequent filling [5]. Khayrutdinov et al. used the FLAC 3D modeling software and changed the filling strength to control the stress–strain behavior of rock mass, to reduce the impact of underground mining on the surface, and to improve mining safety [6,7]. Li et al. proposed a dynamic cross layout model based on IDZs, which could dynamically adjust the sublevel height and drift spacing according to the ore-rock bulk flow parameters, economic indicators, ore body occurrence conditions, drilling machine, etc. [8]. Taking the Chengchao Iron Mine as the engineering background, Tan et al. used the method of combining a theoretical calculation, a numerical simulation, and a physical similarity experiment to sublevel height, production drift spacing, and drawing space [9].

The design of stope structural parameters is a complex nonlinear problem that usually involves multiple decision variables such as rock mechanics, mining ground pressure, loss, and the dilution index. It is difficult to express the relationships among the parameter variables by using exact mathematical and mechanical expressions, and correlations among the variable parameters are weak. With continuously increasing mining depth and the continuous expansion of mining scale, the geological environment and stress conditions of underground mining sites become more complex [10], and as a result the existing methods for designing stope structural parameters are no longer applicable due to their limitations, as summarized in Table 1.

**Table 1.** Design methods for designing stope structural parameters.

| Method | Definition | Limitations |
|---|---|---|
| Engineering analogy method | According to the existing mine data, it is applied to similar engineering objects, and then the corresponding stope structure parameters of the research object are obtained. | With continually increasing underground mining conditions, it becomes more difficult to find underground mining structures under the same mining conditions. |
| Analytical method | Taking some mines as the research object, and aiming at the specific mining conditions and environment of mines, the formula calculation method of structural parameter design of underground stope is constructed by using mechanical theory. | The structure of deep rock mass is complex, uncertain, and fuzzy. Using the ideal mathematical model may have good results in a specific mine, and can not reasonably and comprehensively describe all situations. |
| Numerical analysis method | Using numerical simulation software to establish the geometric model of the stope, and its excavation process is simulated to comprehensively compare and analyze the stress, displacement, and plastic zone distribution of the stope after excavation, and therefore, to obtain better parameters of the stope. | Optimization is carried out on the given stope structural parameters, and there is no method to determine the initial structural parameters. At the same time, the optimization analysis process of the modeling cycle is very cumbersome and heavy workload. |
| Comprehensive analysis method | By studying the relevant factors to determine the stope structural parameters, the corresponding evaluation system model or mathematical formula is established by using some emerging Sciences (neural network, genetic algorithm, and artificial intelligence) to optimize the stope structural parameters. | The calculation results are based on the size of sample data and the means and methods used in modeling, and the results of different methods are quite different. |

Deep stope is a special mining technical condition with strong mining disturbance under high stress. When designing stope structural parameters, we should consider the occurrence of ore body, geological structure, and rock mass quality as well as fully analyze the impact of mining stress induced by deep mining on surrounding rock stability. In particular, it is necessary to break through the stope structure design based on the "experience method" and the "engineering analogy method" and change the design method for stope structure parameters based on deep mining ground pressure response. The design method for stope structural parameters that is proposed in this paper is based on rock mass quality classification, and the structural parameters of deep stope are designed by using an improved critical span graph and an improved stability graph. This method fully considers the key influencing factors such as the occurrence of mining ore body, joint development degree, and mining stress, to ensure the stability of ore and rock in the mining process and to achieve the purpose of safe and efficient production.

## 2. Materials and Methods

### 2.1. Design Principle of Stope Structural Parameters

From the actual production conditions of mines at home and abroad, on the one hand, the stope structure parameters play a decisive role in the stability of the stope, and on the other hand, they also affect the economic benefits of mining [11,12], as shown in Figure 1.

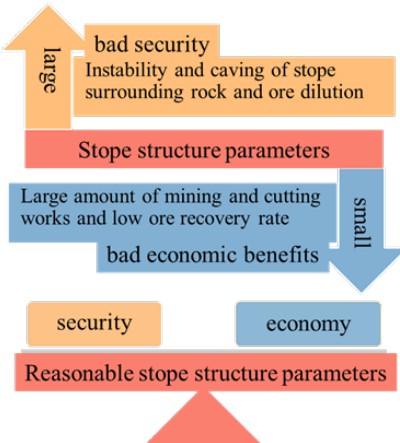

**Figure 1.** Influence of stope structural parameters on safety and the economy.

When the structural parameters of the stope are too large, it leads to instability and caving of ore and rock, increases the loss and dilution, and makes it impossible to operate normally and safely. When the stope structural parameters are too small, it leads to large mining and cutting quantities and low ore recovery, which reduces the economic benefits. Therefore, optimization of stope structure parameters should be considered from two aspects, i.e., safety production and economic benefit. The relationship between these two aspects should be balanced in the design, and the stope structure parameters should be increased as much as possible on the premise of ensuring safety in order to improve economic benefit.

### 2.2. Design Process of Stope Structural Parameters

The design of structural parameters of deep stope is related to the occurrence of a mining ore body, the degree of joint development, in situ stress, and other natural factors, as well as the selected mining method, mining sequence, and ground pressure control method. The underground stope can be roughly seen as a cuboid, which is composed of three elements: length, width, and height; its model is shown in Figure 2.

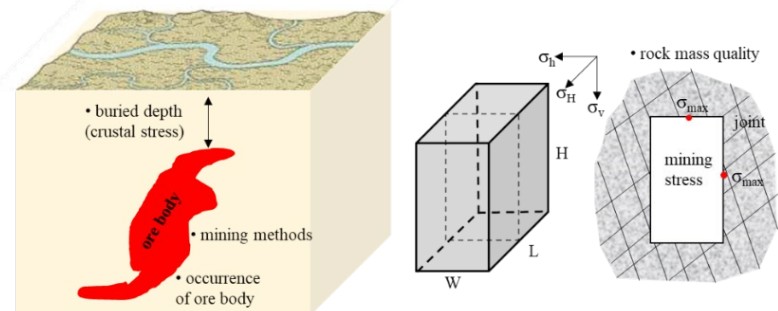

**Figure 2.** Mechanical model of underground stope.

The specific process of stope structure parameter design is shown in Figure 3. The general idea is based on the technical and economic conditions of deposit mining, and a critical span graph and an improved Mathews stability graph are used as tools to incorpo-

rate into the stope structure parameter design the factors affecting stope stability such as rock mass quality, joint occurrence, and mining stress.

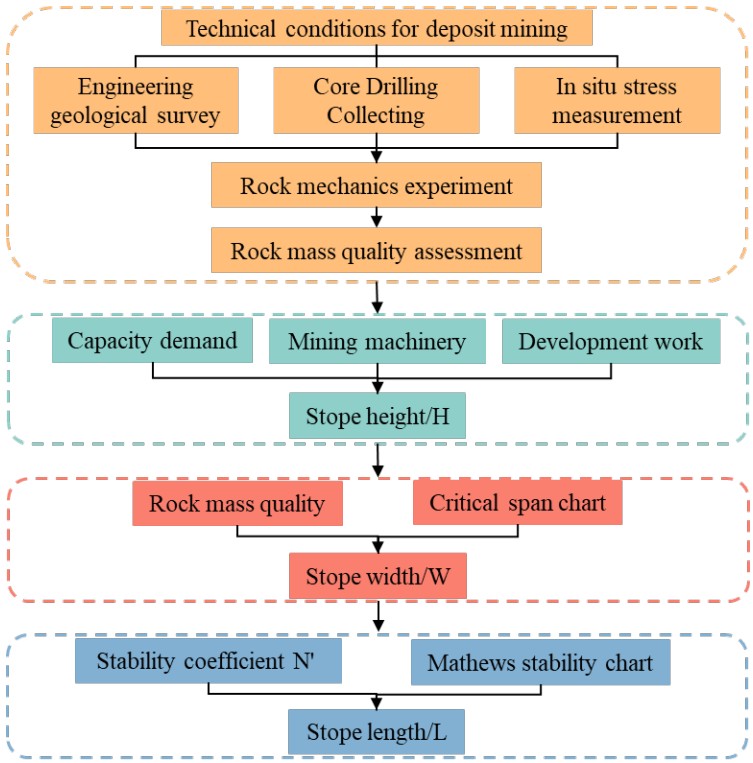

**Figure 3.** Design process of stope structural parameters.

### 2.3. Critical Span Graph

The rock mass rating (RMR) critical span graph has been compiled by experts and scholars at home and abroad based on a large amount of data measurements [13,14]. Since the RMR critical span graph was developed by Lang at the University of British Columbia in 1994 and continuously modified by experts and scholars, it has become a fast and convenient tool for estimating the maximum span that can maintain the stability of underground engineering according to the RMR rock mass classification score [15]. Brady et al. updated the RMR critical span graph in 2003, as shown in Figure 4.

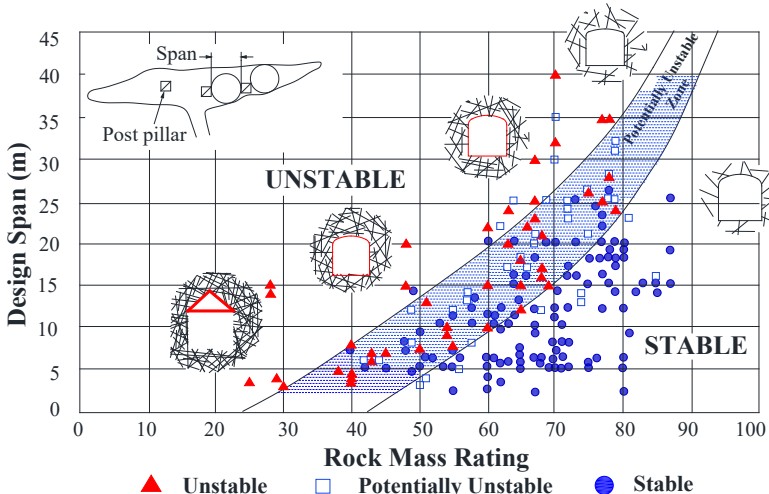

**Figure 4.** RMR critical span graph.

According to the results of rock mass quality classification and based on the RMR critical span graph, the maximum span in which stability can be maintained by underground rock mass excavation can be determined. The RMR critical span graph is used to determine the width of the stope in the parameter design of the stope structure.

*2.4. Improved Mathews Stability Graph*

The Mathews graph method for evaluating stope stability was first proposed in 1980 [16]. Since then, a large number of researchers have collected new data from various mining depths and rock mass conditions, extended this method, and verified its effectiveness.

The original stability graph was only based on 26 cases. After decades of expansion and improvement, the style of the Mathews stability graph has been changed to be applicable to a wider variety of stope sizes, rock mass conditions, and mining methods [17,18].

The original Mathews stability graph contained three different areas, namely, stable area, unstable area, and collapse area, as shown in Figure 5a [19]. Potvin collected more mine data in 1988. He divided the area of the stability graph into stability area and collapse area, as shown in Figure 5b [20]. Nickson and Hadjigeorgiou improved the stability graph of Potvin in 1992 and 1995, adding more cases of supported and unsupported stopes. The improved stability graph is shown in Figure 5c [21,22]. Stewart and Forsyth readjusted the Mathews stability graph in 1995. They subdivided the graph area into four parts through three transition zones, namely stability zone, failure zone, serious failure zone, and collapse zone, as shown in Figure 5d [23]. Based on the extended stability database, Mawdesley used logistic regression analysis to determine the stability boundary and collapse boundary of the extended Mathews stability graph, as shown in Figure 5e [24].

The original intention of the stability graph was to analyze the stability of underground engineering. With the continuous improvement and development by experts and scholars, the Mathews stability graph can also be used in the design of stope structural parameters, but the following problems will be encountered in the design of deep stope structural parameters:

- The quality of deep rock mass is poor, but the existing data $N'$ is concentrated around 100 and is applicable to the situation of good rock mass quality evaluation;
- The values of different mining methods corresponding to the zoning curve are not clear;
- The stability coefficient $N'$ has a large span (0.1~1000) and uneven distribution, inaccurate value, and large error.
- To solve the above problems, the Mathews stability graph is improved as follows:
- When collecting foreign mine data, we focus on sorting and classifying the data with $N'$ less than 100, to increase the degree of data concentration, to increase its applicability when the rock mass evaluation is poor, and to improve the reliability of zoning;
- The collected data are divided into two categories, i.e., unsupported and supported, to make zoning more accurate and adapt to different types of mining methods;
- The coordinate axes are evenly distributed, and the zoning curve is fitted to eliminate the value error.

According to the rule that the hydraulic radius increases with an increase in the stability coefficient $N'$, a linear function or exponential function is selected as the partition function. Research has shown that an exponential function has a better fitting effect than a linear function. The 213 groups of unsupported data and 77 groups of supported data collected are divided by the exponential function, the partition curve is adjusted according to the characteristic points, and the partition function is added. The improved stability graph is shown in Figure 6. The maximum hydraulic radius allowed by different mining methods is determined according to the stability probability of each partition, as shown in Figure 6a,b.

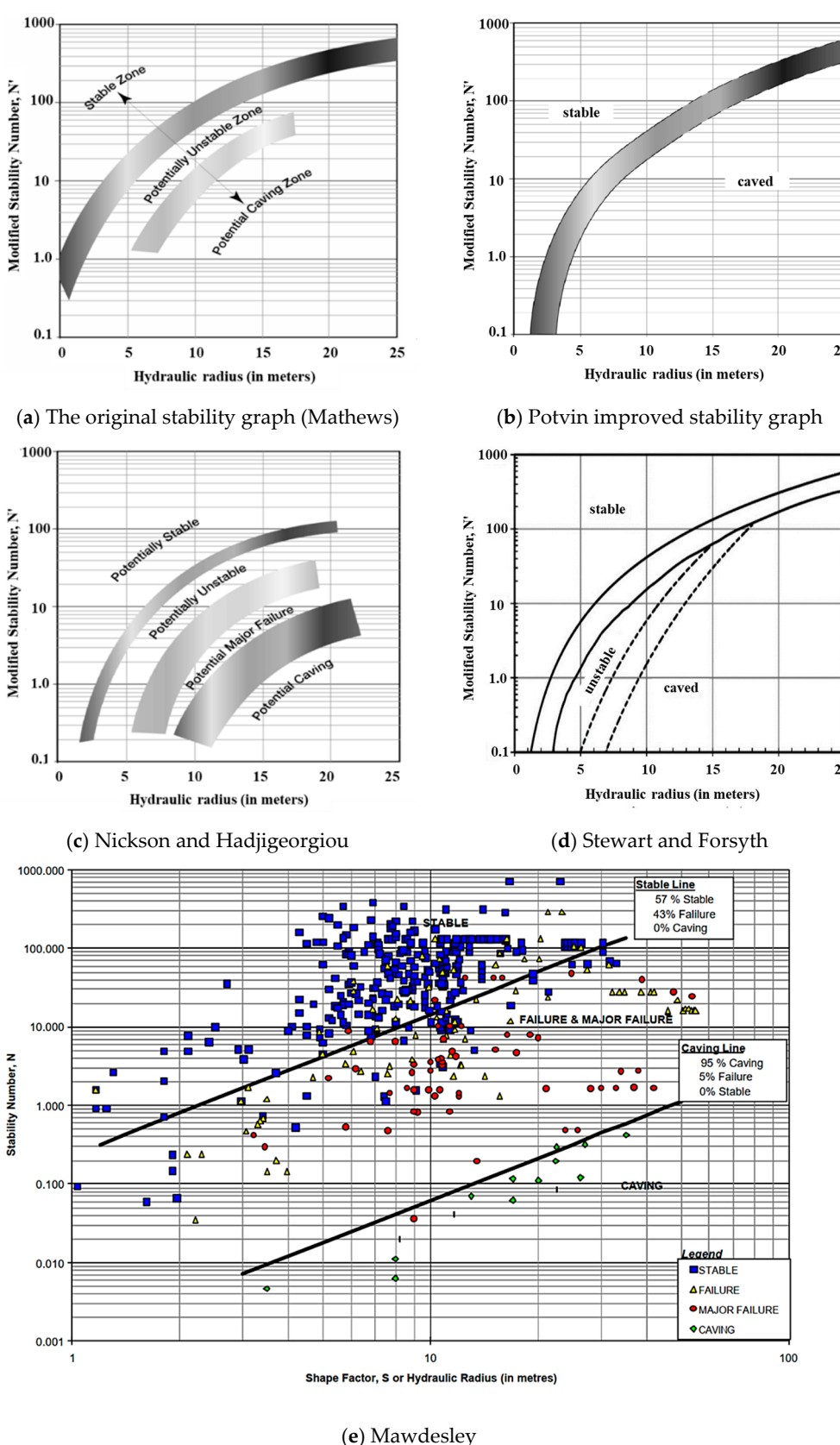

(**a**) The original stability graph (Mathews)　　　(**b**) Potvin improved stability graph

(**c**) Nickson and Hadjigeorgiou　　　　　　(**d**) Stewart and Forsyth

(**e**) Mawdesley

**Figure 5.** Stability graph development process. (**a**) The original stability graph (Mathews); (**b**) Potvin improved stability graph; (**c**) Nickson and Hadjigeorgiou improved stability graph; (**d**) Stewart and Forsyth improved stability graph; (**e**) Mawdesley improved stability graph.

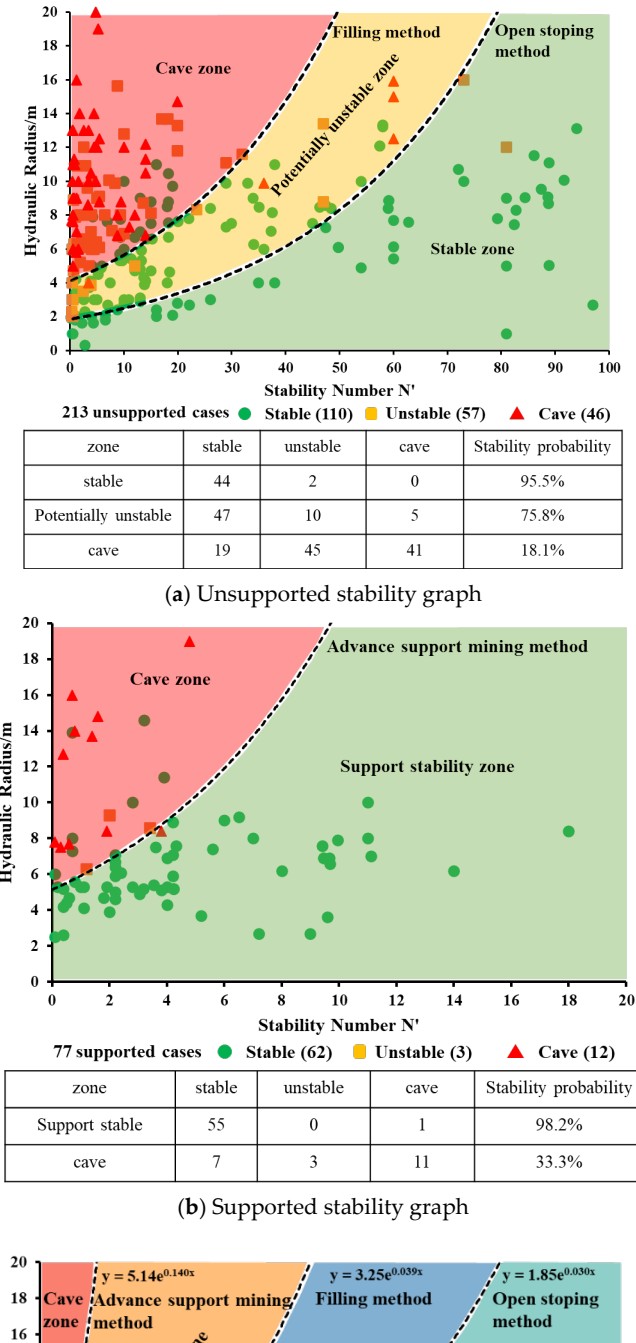

(**a**) Unsupported stability graph

(**b**) Supported stability graph

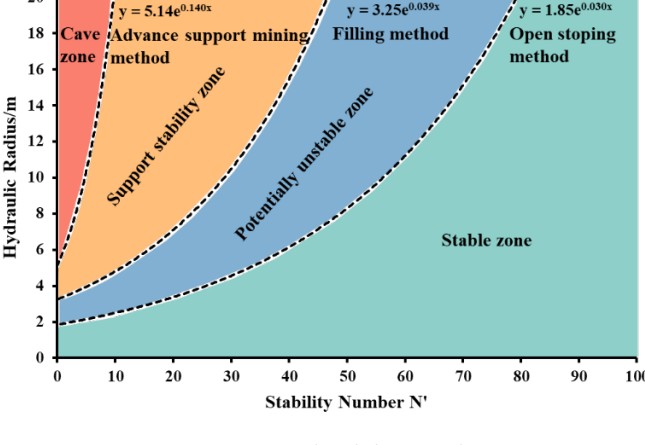

(**c**) Improved stability graph

**Figure 6.** Improved stability graph. (**a**) Unsupported stability graph; (**b**) Supported stability graph; (**c**) Improved stability graph.

The data in Figure 6a,b is integrated to form a complete stability graph, as shown in Figure 6c. The boundary between the stable zone and the potentially unstable zone can be used as the maximum allowable hydraulic radius of the open stoping method. The boundary between the potentially unstable zone and the support stability zone can be used as the maximum allowable hydraulic radius for filling mining. The boundary between the support stability zone and the cave zone can be used as the maximum allowable hydraulic radius of the advance support mining method.

## 3. Case Study

### 3.1. Project Profile

The Sanshandao gold mine is located in Laizhou City, Shandong Province. The ore body mainly occurs in the Sanshandao fault zone. The lithology is mainly pyrite sericitized cataclastic rock and sericitized granite, and the rock mass stability is relatively poor [25]. The three-dimensional model of the ore body is shown in Figure 7. The average dip angle of the ore body is 50° (Figure 7a), and the design stope is located in the middle section from −915 m to approximately −960 m, with an average thickness of about 45 m (Figure 7b). Therefore, it belongs to an inclined thick large ore body.

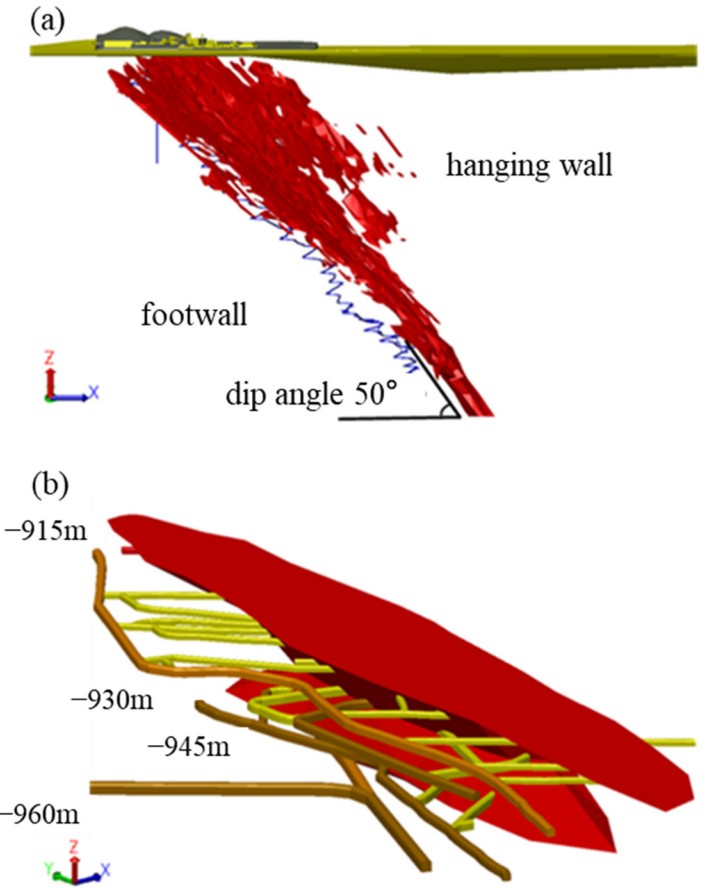

**Figure 7.** Three-dimensional model of ore body. (**a**) Overall three-dimensional model of ore body; (**b**) Three dimensional model of −915m ~ −960m middle section ore body.

In order to ensure the safety and efficiency of mining, it is proposed to adopt the sublevel open stope and subsequent filling mining method. The three views of the mining method are shown in Figure 8.

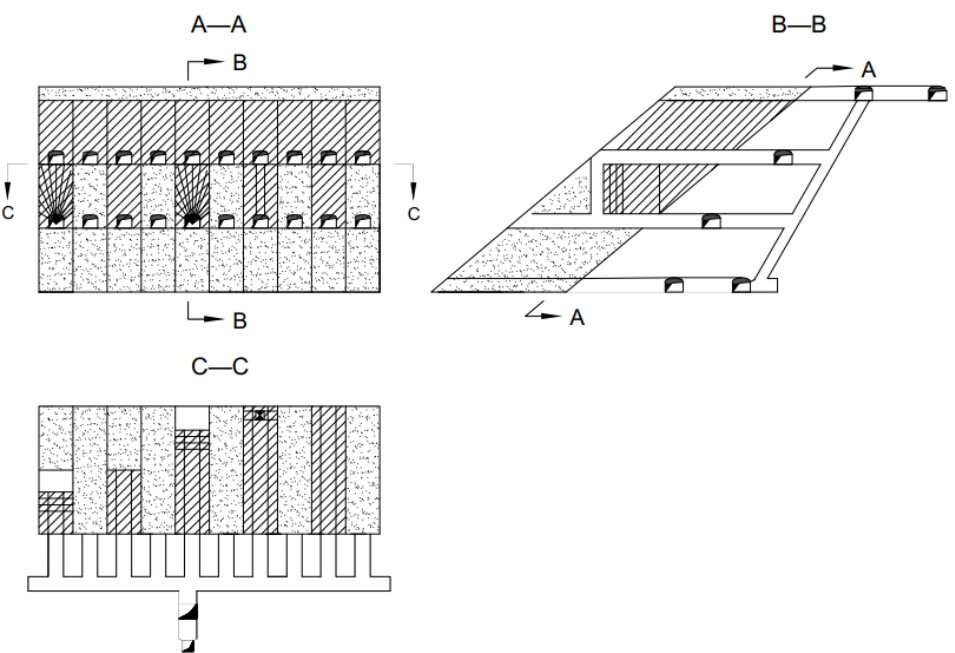

**Figure 8.** Sublevel open stope and backfilling.

The current stage height of the mine is 45 m. According to the existing development project, three sections are set for mining, and the stope height is determined to be 15 m. Then, the limit span (width) of stope stability is determined with the help of the RMR stability graph; on this basis, the stope length is designed by using the improved stability graph.

### 3.2. Rock Mass Quality Estimation

Barton rock mass quality classification (Q) and rock mass geomechanics classification (i.e., RMR) are common methods for quantitative evaluation of rock mass quality grade [14,26]. Before classification, the rock quality index (RQD), joint conditions, uniaxial compressive strength of intact rock, in situ stress conditions, and other evaluation indexes are collected.

The rock quality index (RQD) determines the range of core belonging to the ore body according to the spatial position relationship between exploration drilling and ore body as shown in Figure 9, and calculates the ratio of the total length of the part exceeding 10 cm in the core to the total length of the core. Finally, the RQD value is 23.92.

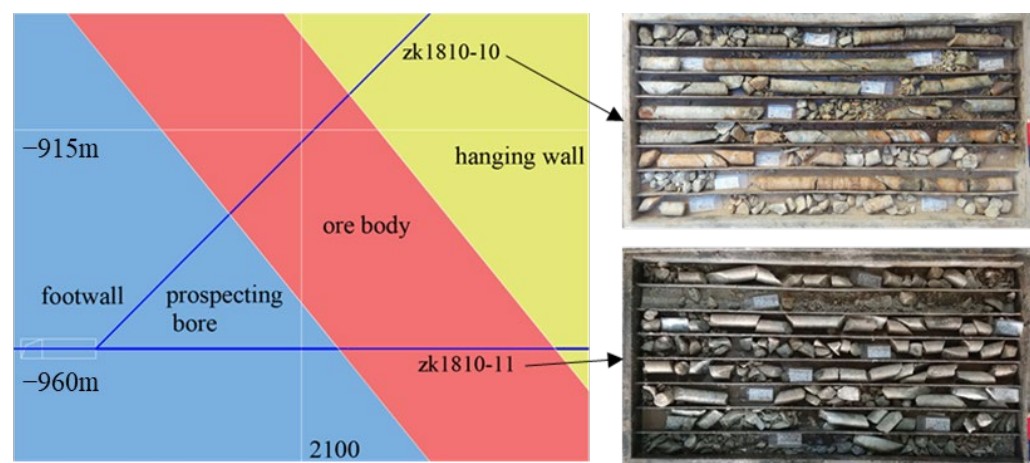

**Figure 9.** RQD calculation from exploration drill core.

The survey line method is used to collect the occurrence information of joints in the exploration roadway in the middle section of −960 m. The occurrence of joints is grouped by Dips software and the occurrence of dominant joints is determined, as shown in Figure 10. The joints are well developed and penetrating, with an opening of less than 1 mm. The roughness of the joint surface is general, partially filled with mud, slightly weathered, and wet.

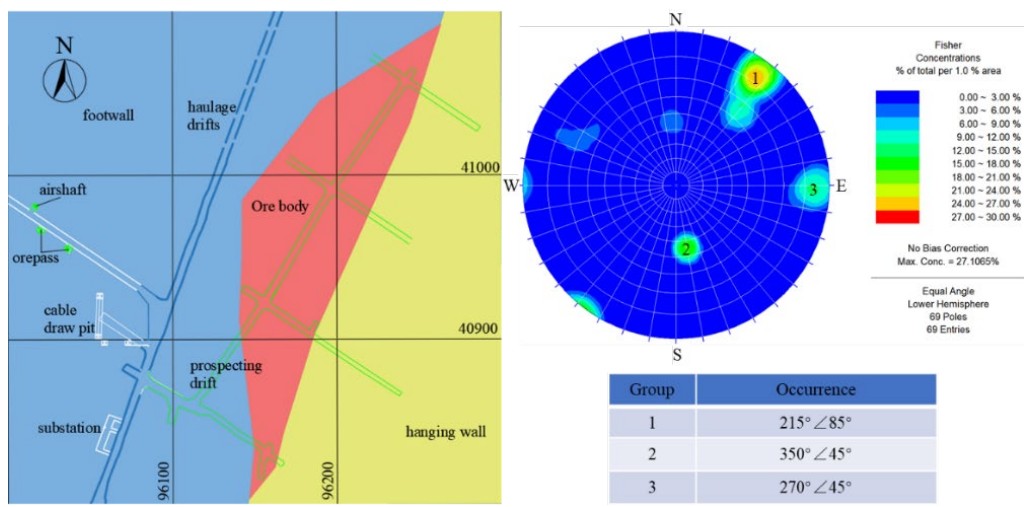

**Figure 10.** Survey joint occurrence in prospecting drift and determination of dominant joints with Dips software.

Rock samples were selected for the rock mechanics experiment, and the uniaxial compressive strength of intact rock was measured to be 86.74 MPa. The Q and RMR rock mass quality evaluation methods were used to evaluate the ore and rock quality, as shown in Tables 2 and 3.

**Table 2.** Quality classification of Barton rock mass.

| Evaluating Indicator | RQD | $J_n$ | $J_r$ | $J_a$ | $J_w$ | SRF | Grade | Rating | Description |
|---|---|---|---|---|---|---|---|---|---|
| Score | 23.92 | 6 | 1.5 | 3 | 1 | 2 | 1.00 | IV | poor |

**Table 3.** Rock mass geomechanics classification (i.e., RMR).

| Evaluating Indicator | Uniaxial Compressive Strength | RQD | Joint Spacing | Joint Surface Condition | Groundwater | Joint Occurrence | Grade | Rating | Description |
|---|---|---|---|---|---|---|---|---|---|
| Score | 7 | 3 | 15 | 15 | 10 | −10 | 40 | IV | poor |

### 3.3. Stope Structure Parameter Design

The RMR critical span graph is used to determine the maximum allowable span (width) of the stope. When the RMR score is 40, the maximum unsupported span is 8 m, and therefore, the width of the stope is 8 m.

According to the above-determined stope width of 8 m and stope height of 15 m, and combined with the indoor rock mechanics experiment and field joint investigation results, the stability coefficient N′ of stope roof and two walls is calculated.

The stability number N′ is calculated as follows [27]:

$$N' = Q' \times A \times B \times C \tag{1}$$

where Q′ is the index of rock mass quality, which is the modified Q system classification method; A incorporates the effects of mining stress; B is the joint occurrence adjustment coefficient; C is the gravity adjustment coefficient.

$Q'$ can be represented by the following equation:

$$Q' = \frac{RQD}{J_n} \times \frac{J_r}{J_a} \qquad (2)$$

where RQD is the rock quality designation, $J_n$ is the joint set number, $J_r$ is the joint roughness number, and $J_a$ is the joint alteration number.

### 3.3.1. A Is the Influence Coefficient of Mining Stress

The value of A is determined by two parameters: mining stress and uniaxial compressive strength of intact rock. The calculation formula are as follows:

$$\sigma_c/\sigma_1 < 2; \ A = 0 \qquad (3)$$

$$2 \leq \sigma_c/\sigma_1 \leq 10; \ A = 0.1125(\sigma_c/\sigma_1) - 0.125 \qquad (4)$$

$$\sigma_c/\sigma_1 > 10; \ A = 1.0 \qquad (5)$$

where $\sigma_c$ is the uniaxial compressive strength of intact rock and $\sigma_1$ is the mining-induced stress.

The mining-induced stress ($\sigma_1$) required for calculating A value is calculated using the RS2 numerical simulation software, and the elastic Mohr Coulomb constitutive model is adopted. The displacement boundary condition is adopted, the horizontal principal stress is 40.7 MPa, and the vertical stress is 25.1 MPa. The calculation results are shown in Figure 11, and the calculation results of A value are shown in Table 3.

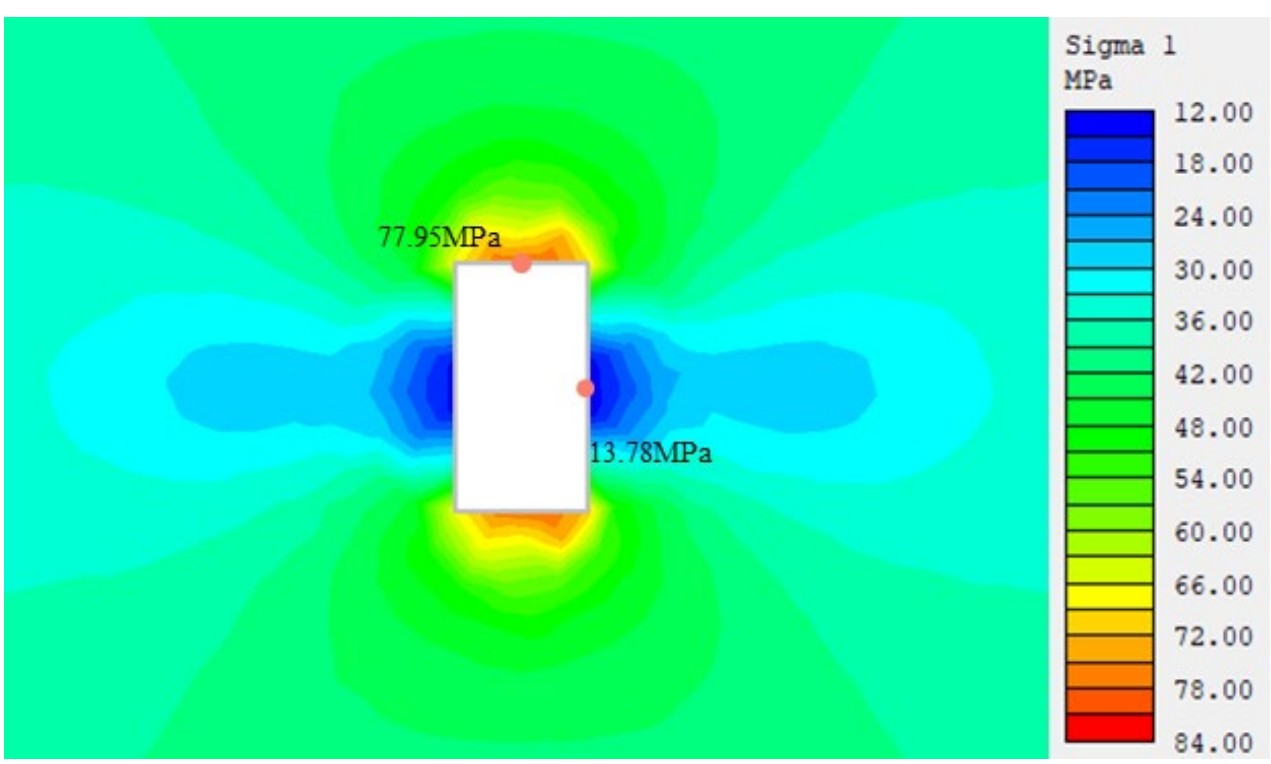

**Figure 11.** Mining-induced stress solution results.

### 3.3.2. B Is the Joint Occurrence Adjustment Coefficient

B is the joint orientation factor. According to the statistical results of joint occurrence, the main dip angle is 45°. Combined with Figure 12, the B value is 0.5.

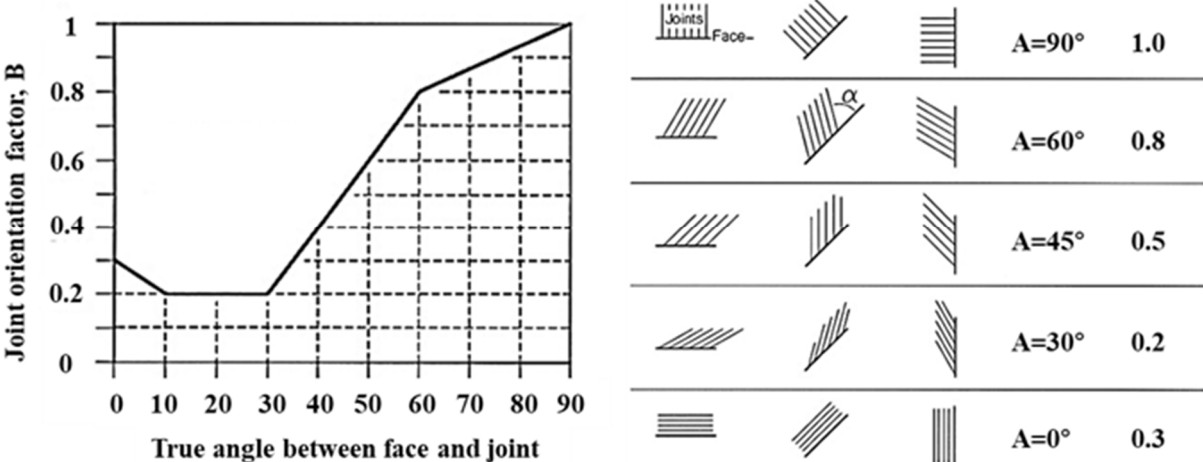

**Figure 12.** Joint occurrence adjustment coefficient.

### 3.3.3. C Is the Gravity Adjustment Coefficient

The failure modes of stope roof and two sides under the action of gravity are mainly considered, such as stope roof caving, slope slicing, slope sinking, and rock mass structural plane sliding. Potvin (1988) believed that gravity-induced failure and spalling failure depended on the dip angle of the mining surface, and the two-slope sinking and sliding failure mainly depended on the dip angle of the stable joint controlling the ore. The value of C can be estimated by the following empirical formula:

$$C = 8 - 6\cos\alpha \tag{6}$$

For stope roof, $\alpha$ is 0° and C value is 2; for stope wall, $\alpha$ is 90° and C value is 8.

According to the rock mechanics experiment, the joint occurrence investigation results, and empirical formula, the calculation results of stability coefficient N′ is shown in Table 4.

**Table 4.** Calculation statistics of stability coefficient N′.

| Position | Q′ | A | B | C | N′ |
|---|---|---|---|---|---|
| Roof | 1.99 | 0.1 | 0.5 | 2 | 0.19 |
| Wall | 1.99 | 0.58 | 0.5 | 8 | 4.61 |

The improved stability graph (Figure 6) is used to determine the hydraulic radius of the roof and two walls, and then the stope length is calculated, according to Formulas (7)–(9), and the results are listed in Table 5:

$$L_{roof} = \frac{2HR_{roof}W}{W - 2HR_{roof}} \tag{7}$$

$$L_{wall} = \frac{2HR_{wall}H}{H - 2HR_{wall}} \tag{8}$$

Stope length L

$$L = \min\left(L_{roof}, L_{wall}\right) \tag{9}$$

where $L_{roof}$ is the maximum allowable length of stope roof; $HR_{roof}$ is the maximum allowable hydraulic radius of stope roof; $L_{wall}$ is the maximum allowable length of stope wall; $HR_{wall}$ is the maximum allowable hydraulic radius of stope wall; $W$ is stope width; $H$ is stope height; $L$ is stope length.

**Table 5.** Calculation statistics of stope length.

| Stope Height/m | Stope Width/m | Position | Hydraulic Radius/m | Stope Length/m |
|---|---|---|---|---|
| 15 | 8 | roof | 3.27 | 36.09 |
|  |  | wall | 3.89 | 16.16 |

According to the calculation results in Table 5, when the stope width is 8 m and the stope height is 15 m, the stope length is restricted by the stability of the two walls, and the maximum length is 16.16 m. The average thickness of the ore body is 45 m, which can be mined in three times, and the length of each time is 15 m. Finally, the stope structure parameter is determined as 8 m × 15 m × 15 m.

## 4. Conclusions

The design of the structural parameters of deep stope is an optimization problem. On the premise of ensuring production safety, larger structural parameters of the stope should be selected to maximize the economic benefits of mines.

Based on the stability graph, in this paper, we outline the design process of stope structural parameters, and improve the stability graph. Although the application scope of the improved stability graph has been reduced, its data concentration is improved, the zoning is more accurate, and it is more suitable for the deterioration of the quality of deep rock mass. In addition, each zone corresponds to the corresponding mining method, which is more targeted in the design of stope structure parameters.

At the same time, taking the stope structure parameter design of −960 m middle section of the Sanshandao Gold Mine as an example, in this paper, we expound the design process of stope structure parameters in detail, which has specific guiding significance for the design of stope structure parameters.

There are many factors affecting stope stability. In this paper, for the design process of stope structure parameters, we considered the key elements such as rock mass quality, joint occurrence, and mining stress; however, we did not discuss the influence of blasting vibration on stope stability. At the same time, the determination of stope self-stability time will be the next research direction.

**Author Contributions:** Writing—original draft preparation and editing, investigation, software, X.Z. (Xin Zhou); funding acquisition, writing—review, X.Z. (Xingdong Zhao); translation, X.Z. (Xingdong Zhao) and X.Z. (Xin Zhou). All authors have read and agreed to the published version of the manuscript.

**Funding:** Project U1806208 supported by the National Natural Science Foundation-Shandong Jointed Fundation of China; projects 52130403 supported by the Key Program of National Natural Science Foundation of China; and project N2001033 supported by the Fundamental Scientific Research Business Expenses of Central Universities.

**Data Availability Statement:** The data presented in this study are available on request from the corresponding author.

**Conflicts of Interest:** The authors declare no conflict of interest.

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
