# Peer review of "Design Method and Application of Stope Structure Parameters in Deep Metal Mines Based on an Improved Stability Graph"

_minerals, doi:10.3390/min13010002_

Round 1
Reviewer 1 Report
The topic of the manuscript is very interesting.
My comments are:
1. equation 2, rock mass quality index (according to Barton) is not complete. Jw/SRF is missing.
2. English language has to be improved.
Author Response
Thank you for your suggestions, which have played an important role in improving the level of my paper. I will revise and reply to your suggestions one by one. Please see the attachment.

Reviewer 2 Report
The manuscript "Design Method and Application of Stope Structure Parameters in Deep Metal Mines Based on Improved Stability Graph" by Xing-Dong Zhao and Xin Zhou was submitted for peer review.
I read the submitted manuscript with great interest. The author turned to a very urgent problem: study of the impact of mining operations on the undermined rock mass and the distribution of stress-strain changes. The authors study the impact of mining operations on the changes of stress-strain state of rock mass in deep mines.
A great deal of research has been done by the authors. But the manuscript has several significant flaws that need to be corrected. Correction of the shortcomings listed below must be done to improve the quality of the manuscript, enhance the ease of perception of the presented material and increase the interest of a readers.
1.) From my point of view the title of the manuscript is a bit vague and does not reflect the essence of the research. The authors are solving rather urgent task: the search of optimal parameters of stope for safe mining works and stabilization of the stress-strain state of mass when mining the deposit reserves at great depths. The key in this case is: safety mining operations. Therefore, in my opinion, it is desirable to reflect this in the title. This comment is advisory in nature.
2.) From my point of view, there are very few keywords. Keywords enable the reader to quickly search for the necessary material and enable the author to popularise their research and increase interest and citations. But if this number of keywords satisfies the requirement of the journal, this comment is advisory.
3.) The abstract is not formed correctly. It is very blurry and framed incorrectly. Abstract is a short and concise presentation of a complex study. It seems that the authors have taken certain phrases from the text and thus formed the abstract. The abstract should clearly indicate the purpose of the study, its importance for society (i.e. to characterize the problem), identify the methods and materials of the study, and the conclusions should be clearly and briefly formulated. There is no "starting point" in the abstract, that is, information about previous studies (one sentence is enough). From my point of view, in the abstract, such information begins with the statement: "Previously conducted studies have established that ...".
3.1) It is desirable to avoid narrative text in the abstract.
3.2) Try to use words and phrases: an analysis has been carried out; studied; developed; proposed; established and so on. It is advisable to start sentences in the abstract with these words and phrases.
3.3) At the end of the abstract, it is necessary to indicate the final result obtained by the authors, for example: A model has been developed that allows ...; A dependence has been established which is...; A pattern has been revealed...; An efficient system (technology) has been proposed, and so on.
The abstract should be revised.
4.) The manuscript has a limited reference list (25 references in total). At the same time there is no comprehensive coverage of studies in terms of geography of citations. There are not enough references to international studies in the field. There are references to foreign research papers, but all of them are older than 5 years. The list of references is intended to demonstrate the depth of the author's study of the material, the relevance and interest of their research. A very high percentage of self-citations: 16% - that is a lot (maybe they are namesakes, then it is all good).
4.1.) The depth of study is demonstrated with the number of references - is not enough.
4.2.) Relevance – with the availability of research in recent years – is enough.
4.3.) Interest – with the availability of research by scientists from different countries - is not enough (practically absent).
Since you are publishing your manuscript in an international publication, it is necessary to demonstrate the international relevance and interest of this issue. This can be done by analyzing the studies of scientists from different countries. It is imperative to supplement the list of references with studies of scientists from different countries over the past 3-5 years to show geographical (general/global) interest and relevance.
The List of References needs to be revised.
5.) From my point of view, there is a slight dissonance when reading the manuscript. The authors state in the title and in the text about "Application of Stope Structure Parameters". However, I have not found any proposed parameters (dimensions) or stope structure in the text. From my point of view, the manuscript needs to be brought into a single vector.
6.) From my point of view, the authors abuse the names of scientists when mentioning the study. A reference [1] is sufficient. If the reader is interested in the name of the researcher, then it is easy to refer to the references list. It is important for the reader to know the essence (main idea) of the disclosed issue, not the name of the researcher.
7.) From my point of view, at the end of the introduction the authors did not quite correctly formulate a brief conclusion of the analytical study of previously performed works. The authors did not summarize their analysis and did not identify unresolved issues. This conclusion should make it possible to characterize the actual question posed, the purpose of the study and the tasks to be solved to achieve this goal. For example: Analyzing the above, it can be noted that ... is a very topical issue. Therefore, the purpose of this study is ... and to achieve this, it is necessary to solve the following tasks: 1); 2); ... Such a conclusion allows the reader to understand the vector of the study, and the authors to correctly formulate the conclusions. It needs to be improved.
8.) When analyzing previous studies, the authors make a number of unforced mistakes or make statements that are not supported by evidence (references). Some statements are very broad and difficult to understand. From my point of view, it is necessary to form more compact sentences, this way you avoid group references.
9.) Considering the comments (3), (4) and (8), I would like to note that the authors have very poorly disclosed the main subject of the study. In recent years, a lot of work has been carried out to study the impact of mining on the rock mass and to recommend measures for reducing strain changes of rock mass and minimising the stress on the surface. Since mining production has a significant effect on the stress-strain behaviour of rock mass, the issues of reducing this influence are very relevant and scientists around the world are trying to minimize it.
For example,
9.1) Adigamov, A.E.; Yudenkov, A.V. Stress-strain behavior model of disturbed rock mass with regard to anisotropy and discontinuities. Mining Inf. Anal. Bull. 2021, 8, 93–103. https://doi.org/10.25018/0236_1493_2021_8_0_93. Unfortunately, this study is published in Russian. But I am confident that if the authors become acquainted with this paper, they will enrich their knowledge.
9.2) Rybak, J.; Khayrutdinov, M.M.; Kuziev, D.A.; Kongar-Syuryun, Ch.B.; Babyr, N.V. Prediction of the geomechanical state of the rock mass when mining salt deposits with stowing. Journal of Mining Institute 2022, 253, https://doi.org/10.31897/PMI.2022.2. This paper analyses the impact of underground mining on the undermined mass. This impact is investigated using FLAC 3D modelling software. Mining operations cause destruction of the mass above, bringing this disturbance to the surface and producing subsidence and sinkholes. Ways to minimise the impact of underground mining on the surface are suggested. From my point of view, this study is suitable to concretise the statement made by the authors in lines 34-36 and allow them to expand their geographical citation.
9.3) Khayrutdinov, A.M.; Kongar-Syuryun, Ch.B.; Kowalik, T.; Tyulyaeva, Yu.S. Stress-strain behavior control in rock mass using different-stregth backfill. Mining Informational and Analytical Bulletin 2020, 2020(10), 42-55. https://doi.org/10.25018/0236-1493-2020-10-0-42-55. In this study, the authors propose to minimize the change in the stress-strain behavior of rock mass using a different-strength backfill. The assessment of the stress-strain behavior of the undermined mass allows to set the stress values boundaries depending on the formation of an artificial mass.
9.4) Kongar-Syuryun, Ch.; Ubysz, A.; Faradzhov, V. Models and algorithms of choice of development technology of deposits when selecting the composition of the backfilling mixture. IOP Conf. Series: Earth Environ. Sci. 2021, 684(1), 012008. https://doi.org/10.1088/1755-1315/684/1/012008. In this study, the authors propose a methodology of selecting a mineral deposit development technology for stress-strain behavior control in rock mass in mining area. The structure of the methodology includes fuzzy models and algorithms that provide processing of large amounts of information and form the significance of environmental factors.
As follows from the presented works (9.1) - (9.4) the authors of the manuscript submitted for review missed a large layer of research related to the impact of mining on the environment. If the authors become familiar with the works presented in (9.1), (9.2), (9.3), (9.4) they will be able to properly form the introduction, enrich their manuscript with international research by scientists from Poland, Czech Republic, Slovenia, Slovakia, Russia, Germany and demonstrate the depth of their material, as well as eliminate the remark (3).
10.) It is necessary to indicate who made figures 1, 2, 7, 8? If this is the author's merit, then it is necessary to indicate: done by the authors; if this is a borrowed drawing, then it is necessary to indicate the source.
11.) The authors use computer modelling to determine the stress-strain state of mass around the stope. However, there are issues they have not disclosed.
11.1) From my point of view, it is necessary to indicate the boundary conditions for modelling;
11.2) It is necessary to explain to the readers why only one stope is modelled. How this situation considers the influence of the workings, the presence or absence of disturbances or voids near this stope. In order to answer this comment, I would recommend to read papers (9.2), (9.3).
12.) Conclusion section is formatted incorrectly. Conclusion – brief summary of the study without repeating the wording given earlier in the manuscript. The authors abuse repetition throughout the manuscript, and Conclusion section is no exception. Such a presentation of the material reduces the ease of perception by the reader of the information presented. Some of the information provided by the authors in Conclusion section has already been reported in Materials and Methods section or is related to Results and Discussion section. This information should be placed in the relevant sections. This information is superfluous for the Conclusion. The mistake of incorrectly forming conclusions is a consequence of the incorrect presentation of the introduction noted by me in remark (7) due to the fact that when writing the introduction, the aims and objectives are not formulated.
Conclusions should briefly characterize the result of the study, for example:
As a result of the study
(1) the dependence of … was obtained.
(2) it was found that ...
(3) and so on.
The conclusion needs to be revised.
Summary: The manuscript is a finished research work. But the corrections are needed. The chosen research topic is relevant. From my point of view, the authors failed to present their research correctly and clearly, which reduced its value and worsened the ease of perception of the material presented.
From my point of view, the manuscript cannot be published in the open press without correction in accordance with my suggestions.
Author Response

(The authors gave the same response as above.)

Reviewer 3 Report
Dear Authors
Scientific comments
It is an interesting work about the methodology of the use of Barton's geomechanical classification, recommendations for tunnel excavation, support applying the RMR System and the correlation between geomechanical classification indices. You must change the use of the word “stope” in the title and text. It should mention attribution of geomechanical classification parameters (parameter and range of values) support and stability period without support.
In 1. Introduction to item 3. Case study, you should describe the type of geomechanical parameters, the ratings used (description and ratings), which appear in Table 2 and 3. You use the N' parameter in Figure 5 (modified stability number, N ´) and in Figure 6 …stability number N´ (uniform the legend and denomination) as in the text. But only later you describe the parameter N´, in item 3.3 stope structure parameter design. It must be before use in the text. You should also talk about Bieniawski's geomechanical classification, before item 3.
Also, in this section (3.3) (line 232) …where Q '- index of rock mass quality, which is the modified Q system classification method; A - incorporates the effects of mining stress; B - joint occurrence adjustment coefficient; C - gravity adjustment coefficient.... For Q´ you present the relationship, but you must also describe the parameters B and C. The parameter A seems to be SRF or ESR (safety index) for competent rock, rock stress problems – explain better.
Figure 10 shows the stereographic projection of the poles of the discontinuities (you should not write joints) and it turns out that it has three families. Before Fig 10 you should describe the stereographic projection and the poles, the definition of families and the rating adjustment for discontinuity orientations, spacing and condition of discontinuities.
Explain the data points projected in Figures 5 and 6, and Hydraulics radius variable used, and line 154.
Line 56 …analyze the impact of mining stress induced by deep mining on surrounding rock stability… explain and implement.
There is no presentation and explanation of Figure 12 in the text and before it.
The bibliography is adequate.
Editorial comments
Table 1 separate the different Methods using horizontal lines
Line 73 to 80, format font size
The subtitles must be on the same page, e.g. Figure 5 and 6
Line 211, Fig.10(?)
Line 239, …The value of a… A
Author Response

(The authors gave the same response as above.)

Round 2
Reviewer 2 Report
The manuscript "Design Method and Application of Stope Structure Parameters in Deep Metal Mines Based on Improved Stability Graph" by Xing-Dong Zhao and Xin Zhou was submitted for second review.
As can be seen from the submitted manuscript and the explanatory note to the review, the authors did a lot of work to make changes in accordance with the comments.
The revised manuscript is a completed scientific study on a highly relevant topic: study of the impact of mining operations on the undermined rock mass and the distribution of stress-strain changes. The revised version of the manuscript, in my opinion, fully satisfies the requirements of a scientific article and can be published in the open press.